# The Potential for New Donkey Farming Systems to Supply the Growing Demand for Hides

**DOI:** 10.3390/ani10040718

**Published:** 2020-04-20

**Authors:** Richard Bennett, Simone Pfuderer

**Affiliations:** School of Agriculture, Policy and Development, University of Reading, Reading RG6 6AR, UK; s.pfuderer@reading.ac.uk

**Keywords:** ejiao, donkey hides, donkey populations, systems modelling

## Abstract

**Simple Summary:**

The demand for donkey hides for gelatin used in Traditional Chinese Medicine, called ejiao, has increased greatly over recent years, resulting in high and rapidly increasing prices. This has put pressure on donkey populations globally and led to theft and illegal trade in donkeys, resulting in concerns for donkey welfare and the livelihoods of those who rely on donkeys. New donkey farming systems have been set up in China in response. We model how quickly these systems may be able to meet the demand for donkey hides. Results show that it will take at least 10–15 years or more for new donkey farming systems to be able to supply the demand for hides. This means that prices of donkeys and donkey hides will continue to increase with continued thefts and illegal trade. This will have further negative impacts on donkey welfare and on the livelihoods of those who rely on donkeys. Chinese ejiao producers will continue to try to source donkey hides from around the world, putting even greater pressure on donkey populations globally.

**Abstract:**

The demand for donkey hides for ejiao, a Traditional Chinese Medicine, has resulted in rapidly increasing prices for donkey hides and donkeys. This has put pressure on donkey populations globally and has implications for donkey welfare and the livelihoods of those who rely on donkeys as working animals. The aim of the research was to explore the feasibility of setting up new donkey farming systems to supply the rising demand for ejiao using a system dynamics model of donkey production. Results show that the size of the initial female breeding herd, reproductive performance, age of reproduction, percentage of female births and average breeding life of donkeys are key variables affecting the time to build up the donkey population to supply the demand for hides, which will be at least ten to fifteen years. The implications of this are: (i) prices for donkey hides will continue to increase, (ii) companies producing ejiao will use other ingredients, (iii) China will continue to source donkey hides from around the world, and (iv) there will be continued theft and illegal trade of donkeys and concerns for rural households reliant on donkeys for their livelihoods and adverse impacts on donkey welfare.

## 1. Introduction

Globally, donkey products, such as milk or meat, have generally been secondary to the use of donkeys for traction and transport. Nevertheless, since antiquity, donkey products have been attributed medicinal, rejuvenating and beautifying properties. In China, medicinal and rejuvenating effects have been attributed to a gelatin, called ejiao, produced from donkey hides. Today, ejiao is considered one of the most valuable products within Traditional Chinese Medicine (TCM) and is a very popular health tonic medicine in China [1,2] and for those using TCM in other countries, with rapidly increasing demand and prices over recent years. This is primarily due to increased disposable incomes and promotion of TCM (and ejiao) within China. Clinical uses of ejiao within TCM are to enrich the blood and stop bleeding (it is commonly taken by some women at/around the time of menstruation), improve the immune system and to treat insomnia and dizziness [3,4,5]. Ejiao is also ascribed anti-aging and rejuvenating effects [5,6]. Although there are numerous companies producing ejiao products, the previously state-owned Dong’e Ejiao company in Shandong Province is the market leader and premium market brand for ejiao. This company only uses donkey hides sourced in China and has set up its own large-scale donkey farms to facilitate this. It refers to donkey farming for ejiao as a new source of wealth of benefit to farmers and to the health and well-being of consumers.

Some have attributed the decline in the Chinese donkey population over the last three decades mainly to the increased production of ejiao [7,8,9], although increased industrialization and mechanization have certainly played their part because communities which previously relied on donkey traction now use motorized vehicles and machinery.

The most recent estimate for the size of the global donkey population is 45.8 million donkeys [10]. The main use of donkeys has traditionally been, and still largely is, for traction and transport of goods and people. For this reason, economic development with the mechanization of transport and agriculture has generally had dramatic impacts on donkey populations [11,12,13,14]. In Europe, the donkey population has declined by more than 80 percent since the Second World War. Countries with large donkey populations in the past, such as Greece and Italy, have seen the most dramatic declines of more than 95 percent [11]. In China, between 1990 and 2018, when the Chinese economy grew at an extraordinary pace, the donkey population decreased by 77 percent from 11.1 million to 2.5 million donkeys [10,15]. Given the large decline in donkey populations in other countries that have seen strong economic development, it does not seem justified to lay the primary cause of the decline in the China donkey population on the ejiao industry.

It is undeniable though that the strong and increasing demand for ejiao is putting pressure on donkey populations worldwide through increased prices for donkeys [16]. Prices of ejiao, donkey hides and donkeys have increased significantly over the last decade or so in particular. Reliable price information is lacking but all evidence suggests strongly increasing prices. Figure 1 shows prices of donkey hides between 2000 and 2015.

Anecdotal evidence suggests that prices have continued to increase. Hancock and Xueqiao (2018) put the value of a donkey hide in 2018 at 3000 yuan (USD 473) [17], while others report an increase from 20 yuan in the year 2000 to 3000 yuan by the end of 2017 [18]. Bloomberg News [19] report prices of donkey hides of up to 8000 yuan (USD 1160).

This rise in prices has been accompanied by increased efforts by companies in China to source supplies of donkey hides from around the world including from Africa and South America and exploration of the use of feral donkey populations in countries such as Australia. A number of governments have been sufficiently concerned about the impact of rising prices on their donkey populations that they have restricted exports of donkeys or donkey hides to protect them. For example, Burkina Faso, Uganda, Tanzania, Botswana, Niger, Mali and Senegal have put restrictions on the trade of donkeys and/or donkey hides as increased prices for hides have led to reports of widespread thefts of donkeys [18,20]. This loss of donkeys for hides has implications for the livelihoods of the rural poor in Africa and other countries given the importance of donkeys as working animals used for a variety of purposes from cultivation of land to transport of materials and people [21]. In addition, there are concerns for the welfare of donkeys, with many documented stories of the abuse and suffering of donkeys as a result of the trade in donkey hides [20,22]. Thus, although the major demand for ejiao comes from within China, it has far reaching global implications for donkey populations.

In this paper, we focus on the potential to supply donkey hides (not meat, milk or other donkey derived products) from donkey farming systems. New donkey farms for the supply of hides have been set up in China in recent years [23]. It is reported that, in recent years, 15 provinces and 22 cities have announced their intention to provide subsidies to stimulate donkey breeding [17]. Donkeys are relatively costly to breed because of their long gestation period, which is a disincentive to farmers to breed them. A representative from one of the market leaders of ejiao production is said to have indicated that, by 2020, his company will be able to meet their basic needs for hides through their donkey farms [17]. These donkey farming systems to produce hides are unlike any seen before globally in terms of their scale, intensity, technology and level of investment.

The main aim of this paper is twofold. Firstly, we bring together information on donkey reproduction, donkey farming and the ejiao market. Secondly, we use this information in a system dynamics model to assess the potential for donkey hide production from a farming system in China to meet demand in the short and longer term.

System dynamics models help to clarify, understand and simulate the behaviour of complex systems, which is often highly counterintuitive [24]. Livestock systems are complex, with feedback loops and often counterintuitive dynamic impacts. System dynamics modelling has been used in a number of papers to better understand herd dynamics and manage farming systems [25,26,27,28,29,30,31,32]. However, to date, to our knowledge, system dynamics modelling has not been applied to the evaluation of the dynamic behaviour of a donkey farming system.

The primary focus of system dynamics models is not on making precise predictions but on making imprecise, conditional projections of dynamic behaviour [33]. In the case of the donkey farming system, this approach is particularly useful as the dynamic behaviour of the system depends on a number of reproduction and farming parameters that are currently uncertain. In this situation of uncertainty, conditional “what if” projections under different scenarios can be particularly insightful.

## 2. Materials and Methods

### 2.1. Approach

The focus of this paper is on the potential of a donkey farming system to meet the demand for donkey hides (not including meat, milk or other donkey-derived products) from the ejiao market. Information on ejiao production and demand is scarce and mainly comes from the grey literature and media reports [16]. In recent years, information on donkey reproduction has improved due to an increased interest in donkey breeding, mainly in Europe and the Americas. Nevertheless, compared to horses and other domesticated animals, knowledge about the reproductive physiology of donkeys and donkey breeding management is still very limited [34,35,36,37,38,39,40]. In addition, intensive donkey farming is a relatively recent development, mainly for the production of donkey dairy products in Europe and for donkey hides but also meat and dairy in China [23,34,41,42,43,44,45]. To date, intensive donkey farming is still under-researched and knowledge on donkey farming systems is very limited. System dynamics (SD) modelling with its focus on conditional projections of dynamic behaviour rather than on precise predictions is therefore particularly well suited to the study of a donkey farming system to meet the demand for donkey hides for the ejiao industry. See Turner et al. for an overview of SD modelling applied to agricultural and natural resource issues [46].

Two SD models were built and run using Stella^®^ Architect 1.3.1 (isee systems 2017). Monthly time steps were chosen to make the interpretation of changes in parameters such as gestation length or age at slaughter more intuitive (e.g., 13 months rather than 1.08 years). Parameters such as mortality, which are generally defined as annual rates, can be entered into the model as annual rates and are converted into monthly rates within the model. Interfaces are included in the model to facilitate communication of the results to interested stakeholders.

In a first step, a SD model is applied to assess how gestation length, conception rate, abortion rate and the percentage of twins affect the reproductive performance of female donkeys (jennies). Reproductive performance is defined as the average number of foals per jenny per year in a stable population. The model is run over 240 months (i.e., 20 years) to ensure that a steady state is reached. The results of the last 12 month period are presented and discussed. 

In a second step, the output from the reproductive performance model is included in a SD model of a donkey farming system. The objective of the model is to explore the impact of reproductive performance, farming characteristics and hide demand on (a) the time it takes to build up a farming system to meet donkey hide demand for ejiao production, (b) the size of the required breeding population and (c) the total herd size. The model is run over 600 months (i.e., 50 years).

### 2.2. Reproductive Performance Model

In the first step, reproductive performance of a stable population of female donkeys is modelled based on gestation length, conception rate, abortion rate and the percentage of twins. Here, reproductive performance is the average number of foals per jenny per year (Appendix A for the equations of the reproductive performance model and Appendix B
Figure A1 for the user interface of the model). Figure 2 shows the stock and flow diagram of the model.

The population is constant at 1000 jennies (chosen as a convenient reasonably substantial round figure from which outcomes for larger or smaller starting populations can easily be extrapolated and which reflects the size of some enterprises already set up in China). The stock of jennies is divided into pregnant jennies (JP) and non-pregnant jennies (JNP). The system is initialised with 900 pregnant jennies and 100 non-pregnant jennies. Conception (C) is the flow of non-pregnant jennies to pregnant jennies and is determined by the conception rate. The stock of pregnant jennies is a conveyor stock. The time it takes jennies to move through the pregnant jennies conveyor stock is set by the gestation parameter. Abortions (AB) are a leakage from the conveyor stock and are determined by the abortion rate. The outflow of the pregnant jennies conveyor stock is giving birth (GB).
(1)$$JP_{t}=JP_{t-dt}+C_{t}-{GB}_{t}-{AB}_{t}$$

The abortion outflow and the giving birth outflow make up the inflow into the non-pregnant jennies stock in the next period. Annual reproductive performance (RP) is defined as the total number of births (TB) over 12 months divided by the total number of jennies (TJ) in the steady state (i.e., $RP=\left(\sum_{i=1}^{12}TB_{i}\right)/TJ$).

Knowledge about the reproductive physiology of donkeys and donkey breeding management is incomplete. The majority of the published research has been carried out in Europe [11]. Donkey farming in China mainly uses Chinese breeds with the Dezhou breed the preferred, but not the only, breed used by donkey farms that supply the hide market [43,45,47]. Where information is available for Chinese breeds, more weight is given to the values reported for Chinese breeds than to information on other donkey breeds. For each of the four main inputs into the reproduction performance model, low, middle and high parameter values are defined based on the literature, wherever possible.

Gestation lengths have been studied for a number of donkey breeds in recent years [12,34,35,38,48,49,50] but not for Chinese breeds. The reported average gestation length ranges from 353 days [12] to 371 days [48,49,50]. The shortest and longest gestation are reported by Galisteo and Perez-Marin [35] as 331 and 421 days, respectively. Simulations are run with average gestation lengths of 11.5, 12 and 12.5 months, equivalent to 350, 365 and 380 days, respectively.

Reported conception rates within and between different breeding methods (natural breeding, hand breeding, artificial insemination (AI) with fresh, cooled or frozen-thawed semen) vary widely. In a large-scale farming system, the use of AI is likely to play a major role. Outside China, AI for donkeys with frozen-thawed semen has only recently received attention [37,40,51,52,53]. In a review of the use of AI in equids in China between 1935 and 2012, Deng et al. [43] report conception rates between 52.3 percent and 73.2 percent, whilst Chang et al. [54] report conception rates between 76.0 percent and 70.8 percent for timed AI with frozen-thawed semen, which is high compared to studies from outside China. China has a long history of using AI in donkeys going back to the 1940s and AI use has been widespread since the 1950s [43]. Wu [55] reports that using oestrus synchronisation and other techniques, the conception rates for Dezhou donkeys in the National Black Donkey Breeding Center in Shandong, China, have increased from 45 percent in 2013 to 92 percent in 2016.

Based on the literature on Dezhou donkeys, we use a middle value for the conception rate of 0.70, which is at the lower end of the conception rates reported for Dezhou donkeys. We use a conservative value for two main reasons. In the model, after birth of a foal, jennies are immediately subject to insemination again, i.e., during foal heat. Foal heat, the first heat after giving birth, usually occurs in the first two weeks after a jenny has given birth to a foal [51]. In donkeys, foal heat can be used successfully but with slightly lower conception rates than in the following cycles [38,49]. Secondly, the conception rates reported in the literature are mainly based on controlled research studies. It is likely that conception rates in a (less controlled) large-scale farming system will be lower. On the other hand, in the literature, conception rates are often defined by the oestrous cycle, which is approximately 25 days for most donkey breeds studied [39]. The model defines the conception rate on a monthly basis. A low value of 0.60 foals per jenny per year and a high value of 0.80 are used in our simulations.

Information on the abortion rate is scarce. Most studies on donkey reproduction do not report the number of abortions, with a few notable exceptions [12,34,37,49]. In Carluccio et al. [49], one out of 73 pregnancies, i.e., less than two percent, is reported to have ended in abortion. The abortion occurred before day 45 of the pregnancy. Nervo et al. [34] report eight abortions out of 38 pregnancies, i.e., over 20 percent. Abortions occurred between days 236 and 342. Wang et al. [45] report an unusually high number of abortions on an intensive donkey farm in China. Over a period of several months, 61 of approximately 500 pregnant jennies, i.e., approximately twelve percent, aborted their foals between the 200th and 300th day of pregnancy. This was reported as an unusual event, suggesting that abortion rates on the farm are usually substantially lower. Based on the literature, abortion rates are likely to be somewhere between 2 percent and 20 percent. We use 10 percent as the middle value, 5 percent as the low value and 15 percent as the high value for abortion rates.

Little is known about the potential for donkeys to carry viable twins. Multiple ovulations are common in donkeys. The percentage of multiple ovulations reported are between 5 and 70 percent [39]. Yang et al. [56] find that depending on the season, 16 to 34 percent of Dezhou black donkey jennies experience double ovulation. Often double ovulations do not result in viable twin pregnancies [34]. A widespread practice is that if twin embryos are detected, one of them is crushed [49]. By contrast, in the study carried out by Galisteo and Perez-Marin [35], 3 out of the 58 pregnancies were twin pregnancies that were carried to term and resulted in viable foals. It would seem possible that the twin percentage could be increased in an intensive farming system. For sheep in 1990, 101 lambs were weaned per 100 ewes presented for breeding. By 2015 the number of lambs weaned per ewe had increased to 125 [25]. We base the middle value on Galisteo and Perez-Marin [35] assuming 5 out of 100 pregnancies are twins. The low value assumes no twins, i.e., the assumption that twin embryos are crushed. A high value of 10 percent of pregnancies being twin pregnancies is used. This assumption would require advances in viable multiple births similar to those seen in sheep.

### 2.3. The Donkey Farming System Model

In the second step, a model of a farming system is applied to shed light on the likely time frame for developing a large-scale donkey farming system as well as on the herd characteristics of such a system. Figure 3 shows the interface of the model (Appendix C for the equations of the farming systems model and Appendix D
Figure A2 for the model diagram).

In the farming system model, stocks of male and female donkeys are modelled as different sub-models. Each sub-model has four equivalent stocks: (1) a conveyor stock from birth to slaughter maturity (BS), (2) a reservoir stock at slaughter age (SA), (3) a conveyor stock from slaughter age to reproduction age (RA) and (4) a reservoir stock of breeding donkeys (BD).

These stocks are determined by the following relationships.
(2)$${BS}_{t}={BS}_{t-dt}+B_{t}-{SM}_{t}-M_{t}$$
where *B* are births, *SM* are donkeys reaching slaughter maturity and *M* is mortality.
(3)$${SA}_{t}={SA}_{t-dt}+{SM}_{t}-R_{t}-S_{t}$$
where *R* are replacements for breeding and *S* are animals slaughtered.
(4)$${RA}_{t}={RA}_{t-dt}+R_{t}-{MR}_{t}-M_{t}$$
where *MR* are mature replacements
(5)$${BD}_{t}={BD}_{t-dt}+{MR}_{t}-M_{t}-C_{t}$$
where *C* are animals culled at the end of their breeding life.

The model minimises the time it takes to build up a breeding donkey population that can supply a target number of hides on an annual basis. In the build-up phase of the farming systems, all female offspring enter the breeding herd, while most of the male offspring are slaughtered for their hides. The model approximates the jenny requirement for the target hide production and compares this to the replacements already in the pipeline plus the current number of breeding jennies, adjusted for future culls. When the required number of breeding jennies is reached, only jennies required to keep the breeding herd stable enter the breeding herd. The hides of donkeys that are culled from the breeding herd are also used for ejiao production. Donkey hides in the mortality flows are assumed not to count towards hide production.

The variables included in the interface (see Figure 3) are presented in this section.

The target hide variable determines the number of hides per year that the farming system is built to achieve. Production of ejiao is estimated at between five and six thousand tonnes annually [4,18,57]. Information on how much ejiao can be produced from one donkey hide varies widely. Based on these different sources, we estimate that on average one donkey hide produces between 1.25 kg [18] and 2.65 kg (report on Dong’e Ejiao (DEEJ), a large producer of top-quality ejiao on a Chinese platform (in Chinese) http://www.sohu.com/a/129809208_135357) of ejiao. These estimates, together with an estimated annual demand for ejiao production of between five and six thousand tonnes, translate into a donkey hide demand of somewhere between two and five million hides per annum. We run our model with an annual target hide production of 2.0, 3.5 and 5.0 million hides a year.

The farming system starts with an initial female breeding stock. Intensive farming of donkeys is already happening in China [23,44,47]. The latest donkey population estimate is 2.5 million donkeys in China in 2018 [15]. Of these, probably approximately one million are adult female donkeys. It is unknown how many of these are kept as pack and draught animals and how many are part of a donkey farming system. We run simulations with initial female breeding stocks of 50,000, 100,000 and 200,000 jennies, i.e., approximately five, ten and twenty percent of the total adult female donkeys in China.

The reproductive performance values are based on our analysis in step 1. Donkeys enter puberty between 1 and 2 years of age [36,51]. In many research studies, the youngest breeding animals included are two or three years of age [34,35,38,39,53,54,55,56]. The oldest animals in the non-Chinese studies are often between fifteen and twenty years but, for the two Chinese research studies, the oldest jennies were seven and ten years [54,56]. Information on farming systems is scarce but Fanelli et al. [40] studied a donkey dairy farm with jennies aged between three and twenty years old. On the donkey dairy farms studied by Dai at al. [41], the oldest animal was 29 years. We simulate the model with donkeys entering the breeding herd at 24, 30 and 36 months. The literature suggests that donkeys can enjoy a long breeding life, with animals between the ages of fifteen and twenty years regularly used for breeding, but the two Chinese studies do not use animals over ten years of age. Based on this information, the average breeding life from entering the breeding herd until culling is set at 10, 12 and 15 years.

In the build-up phase of a donkey farming system, having more female offspring is desirable. Sperm sexing technology exists and has been applied to and assessed for dairy and beef cattle but more recently also to polo horses [58,59]. In a recent study on sperm sexing in donkeys, Domínguez et al. [59] report that after the sexing procedure, 90 percent of sperm has the X chromosome. Sexing sperm can negatively impact conception rates although the new sexing technique had no impact on conception rates in horses. The authors report that research is under way to assess the impact on conception rates in donkeys. Since the technology is not widely used yet, we simulate the model with the percentage of female foals of 50, 55 and 65 percent.

Those donkeys not retained for breeding are slaughtered for their hides. Wu [55] reports that, in traditional systems, it takes 24 months to raise a donkey for its hide but that the period has been shortened to 16 to 18 months. We simulate the model with slaughter ages of 17, 22 and 24 months.

The model includes two mortality rates. Foal mortality is the percentage of donkeys that die before they reach slaughter maturity. Mortality of breeding donkeys is the percentage of breeding animals that die per year. Mortality of breeding donkeys is separate to and in addition to the cull rate of breeding animals. The cull rate is derived from the average breeding life variable. An average breeding life of ten years translates into an annual cull rate of 10 percent.

Data on mortality are very scarce. Few research papers report mortalities. It is not clear whether the lack of reporting of mortality is due to the fact that none of the donkeys died during the study or whether animals that died were excluded from the studies. Quaresma et al. [13] carried out a survey of donkey owners whose donkeys were registered in a studbook. The authors found that 725 of the 760 donkeys born between 1982 and 2012 and registered in the studbook were alive at the end of 2012. This means that less than five percent had died and suggests an annual mortality rate of substantially below five percent. Data on the donkey population and deaths are available for Ethiopia from the 2017/2018 agricultural survey [60]. The data suggest an annual mortality rate of approximately five percent. The deaths are likely to mainly be of older donkeys. In the farming system, these would be equivalent to the culled animals. Mortality in the farming system model includes deaths that occur before and in addition to the animals culled at the end of their breeding life. The Ethiopian figures therefore also suggest a mortality rate for breeding donkeys well below five percent. We use foal mortality figures of 10, 7 and 5 percent and additionally adult donkey mortality figures of 4, 3 and 2 percent per year.

The model also includes the jenny to jack ratio. This ratio will depend on the breeding method used, with more jacks needed if natural breeding is predominant. We use jenny to jack ratios of 10, 20 and 30 to one in our simulations.

## 3. Results

### 3.1. Reproductive Performance

Table 1 summarises the parameter values and the results for the three reproductive performance scenarios.

The middle scenario results in a reproductive performance of 0.95, or 95 foals being born per 100 jennies in the breeding herd. The reproductive performance of the less favourable scenario is 83 foals per 100 jennies in the breeding herd, i.e., a 13 percent lower reproductive performance than in the middle scenario. For the more favourable scenario, the model indicates that 109 foals are born per 100 jennies in the breeding herd—14 percent higher than in the middle scenario.

Table 2 shows the results of sensitivity analysis. We vary each parameter at a time and set all the other parameters at the values of the middle scenario.

The results show that the twin percentage has the largest impact followed by gestation length. The reproductive performance rates from three main scenarios are used in the farming system model described in the following section.

### 3.2. The Donkey Farming System

We simulate the model for nine different scenarios. We combine the three different target hide settings with the less favourable, middle and more favourable scenarios. In the middle scenario, all parameter values are set to their middle values as reported in Section 2. The middle scenario is most likely to represent what is currently achievable. However, it seems appropriate to re-emphasize here that the aim of this study is not to make precise predictions but to produce conditional projections of dynamic behaviour under “what if” scenarios—what if a donkey farming system could achieve the reproductive and farming performances reported in the literature.

Table 3 summarizes the parameters used in the simulations and the main results.

One important question is how long it might take to set up a farming system that can meet the demand for hides. We report the time in years, not months, to reflect the uncertainty present. We also report the total number of donkeys in the farming system and the number of breeding jennies.

Figure 4 shows that with regard to the time period it takes to reach the target hides, the different farming system performance scenarios have a greater impact than the target hide scenarios.

The target hide output is reached in 20 to 25 years in the middle scenario for all three target hide settings. It takes 38 to 46 years (over 80 percent longer) in the less favourable scenario and it takes 11 to 14 years (between 45 and 50 percent shorter) in the more favourable simulation setting to achieve the three target hide outputs. Comparing the 5.0 million hide scenario with the corresponding 3.5 million hide scenario extends the time taken to produce the target output by between 7 and 11 percent. Comparing the respective 2.0 million target hide and 3.5 million target hide scenarios, the time taken reduces by between 11 and 15 percent. 

Figure 5 shows that in terms of the number of animals required in the farming system to produce the target hides, the number of animals is affected more by the variation in target hides than by the variation in farming system performance.

In the middle scenario for farming system performance, farming systems with 6.5 million donkeys, 11.4 million donkeys and 16.4 million donkeys are required for annual outputs of 2.0 million, 3.5 million and 5.0 donkey hides, respectively. Under the less (more) favourable scenario, the number of donkeys varies from 8.1 (5.8) million for 2.0 million hides to 20.2 (14.6) million for 5.0 million hides.

Table 4 shows the results of sensitivity analysis for the individual parameters. The impact of varying individual parameters on the time it takes to reach the target hides and the number of donkeys in the farming system are presented in percentage terms. The time taken to reach the target hide output is most sensitive to changes in the initial breeding herd and reproductive performance. The donkey numbers are most sensitive to the target hide values and the reproductive performance, with the latter mainly impacting on the number of breeding jennies as would be expected. The reproductive age, the lengths of the average breeding life and the percentage of female births have a substantial impact on the time it takes to reach the target hides but little impact on total donkey number and little or no impact on the number of breeding jennies. The contributions of other variables to changes in the time or donkey numbers are small.

## 4. Discussion

Our simulations suggest that it is likely to take at least ten to fifteen years, if not much longer, to build up a Chinese farming system to supply the current demand for donkey hides from the ejiao industry. Our model results indicate that, for an annual production of 2.0 million donkey hides, the number of donkeys in the donkey farming system needs to be 5.8 million under favourable reproduction and farming conditions, approximately double the current total donkey population in China. For an annual production of 5.0 million donkey hides, our model suggests that 14.6 million donkeys would be required under the favourable farming system performance scenario—or almost six times the current donkey population in China. Of course, in 10–15 years, demand for hides could be significantly higher than currently (e.g., due to increased population and disposable incomes in China), providing an even greater challenge to donkey farming systems to supply the market. Indeed, the demand for the use of TCM (including ejiao) has received a likely boost from the WHO inclusion of a chapter on TCM in its highly influential International Classification of Diseases (11th edition) which helps set the medical agenda globally and may contribute to the acceptance of TCM alongside conventional (western) medicine [9].

The assumptions underlying the more favourable estimates are rather optimistic. Although there are opportunities for better control and more efficient management on large-scale intensive donkey farms, large farms pose management challenges and disease risks [23,47]. For example, Yang et al. [61] report on an equine influenza outbreak on a 300 head donkey farm in Shandong which resulted in 25 percent mortality. Wang et al. [45] study the increase in abortions due to *Salmonella abortus equi* on a farm with over 1000 jennies. These examples show that it may take time to achieve reproductive and other performance levels found in studies carried out on smaller, less-intensive farms. From the above analysis, we can conclude that it is likely to take much longer than a decade for China to meet the demand for donkey hides from a domestic donkey farming system. This has negative implications for donkey welfare in that the economic incentive will continue for theft, illegal trade and trade (including transport and slaughter of donkeys) which gives little or no consideration to the welfare of the donkeys involved (see The Donkey Sanctuary, 2019 for some documented case studies). In the longer term, new large-scale donkey farming systems will have some economic incentive to ensure reasonable levels of donkey welfare and there is an important role for organizations such as the International Office for Animal Health and the Food and Agricultural Organization to agree international health and welfare standards for these systems. There is also an important role for non-governmental animal welfare charities to advise companies setting up such systems in relation to welfare standards and the commercial benefits of good donkey welfare (and there is already evidence of this happening in China).

The system dynamics model developed in this study does not include any economic considerations. The main reason is the lack of information on the costs of donkey production. Further, little reliable information is available on the price that donkey farmers receive for a hide. There are likely to be economic and business challenges to building up donkey farming systems. It takes approximately three years for any revenue from donkey farming (approximately one year of gestation and two years for the foal to reach slaughter weight). As a result, substantial initial investment is needed, with donkey farming only becoming profitable after several years. On this basis, calls for subsidies have been made by ejiao industry representatives as far back as 2010 [7]. The economic incentive for donkey farmers to increase breeding activities could be provided by subsidies or by further increases in the price of donkey hides. The latter would make investment in donkey farming more attractive but would not address the challenge of the long time period required before newly-established farms become profitable.

As already noted, the donkey hide targets in the simulations are subject to some uncertainty. Firstly, reliable figures on ejiao production are not available. Secondly, different sources on how many donkey hides are required to produce one tonne of ejiao give quite different answers. Thirdly, the potential of imports of donkey hides is also uncertain. According to the latest estimates from the Food and Agriculture Organization, there are just over 40 million donkeys outside China [10]. In most countries, the hides of donkeys do not seem to be used upon their death. The most detailed figures on population and deaths are published by the Central Statistical Agency of Ethiopia [60]. In 2017/2018, the total Ethiopian donkey population was estimated at 8.8 million and deaths were estimated at 481 thousand, i.e., approximately 5 percent of the donkeys died in 2017/2018. Using the same ratio for the total population outside China, there would be a natural annual supply of donkey hides of 2.2 million. However, there are significant economic, cultural and governance barriers to the export of donkey hides. Nevertheless, there is some potential for donkey hide trade to contribute to the supply in the short and longer term, but trade is likely to remain relatively small compared to overall demand. Finally, the current use of donkey hides also depends on the size of the counterfeit market. Some sources suggest that up to 40 percent of ejiao is counterfeit [57]. Counterfeit production is a well-known problem and research on identifying genuine product is going on in China [2,3,4,62,63]. Our results assume that the totality of the production of ejiao is genuine, i.e., made from donkey hides only.

There are a number of possible extensions of this study. As noted above, economic factors could negatively impact on the development of a donkey farming system. The model can be extended to include economic information. A second possible extension is to include milk and/or meat production in the model. At least some of the farms in China produce hides, milk and meat [45].

## 5. Conclusions

The system dynamics model of donkey farming for hides has helped to explore the potential for new systems of donkey farming in China (and elsewhere) to supply the growing demand for hides for ejiao production. Reliable information on the likely productive parameters for donkeys bred in a large, intensive production system using the latest technologies is lacking, with some uncertainty surrounding key variables such as reproductive performance, mortality and breeding life. The model has enabled such uncertainties and their effects on the speed with which the donkey population and output of hides can be increased to be explored and assessed. The study has found that it is likely to take at least 10 to 15 years, if not longer, to build up a Chinese farming system to supply the (current) demand for donkey hides for the ejiao industry, depending on the extent to which the productive performance of donkey breeding herds can be maximized. The implications of this are that (i) prices for donkey hides, and for donkeys, are likely to continue to increase over the short and medium term; (ii) there will be a strong incentive for companies producing ejiao products to diversify their products and ingredients to reduce the amount of ejiao and, for example, include gelatin from other species and other ingredients to ‘stretch’ the amount of ejiao available across products; (iii) China will continue to explore non-domestic sources of donkey hides from around the world until China’s domestic supply of donkeys can be increased; and (iv) there will be a continued financial incentive for the theft and illegal trade of donkeys. Concern will also remain about the possible impact of this pressure on donkey populations for low-income rural households reliant on donkeys for their livelihoods and the potential adverse impact on the welfare of donkeys themselves.

There is a need to continue to monitor the market for ejiao and donkey hides and its impact on donkey populations globally. There is also a need to monitor the development of new donkey farming systems that produce hides and to ensure that these systems have satisfactory standards in relation to biosecurity, donkey welfare and environmental impacts.

## Figures and Tables

**Figure 1 animals-10-00718-f001:**
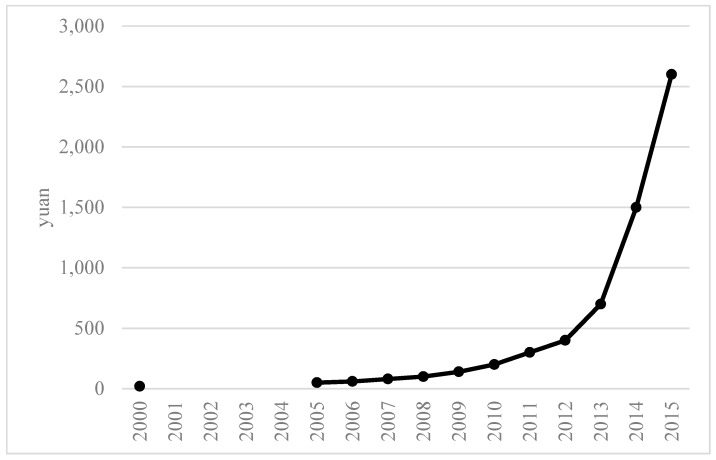
The price of donkey hides in China between 2000 and 2015 (in yuan). (Source: http://www.bestchinanews.com/Domestic/11652.html.).

**Figure 2 animals-10-00718-f002:**
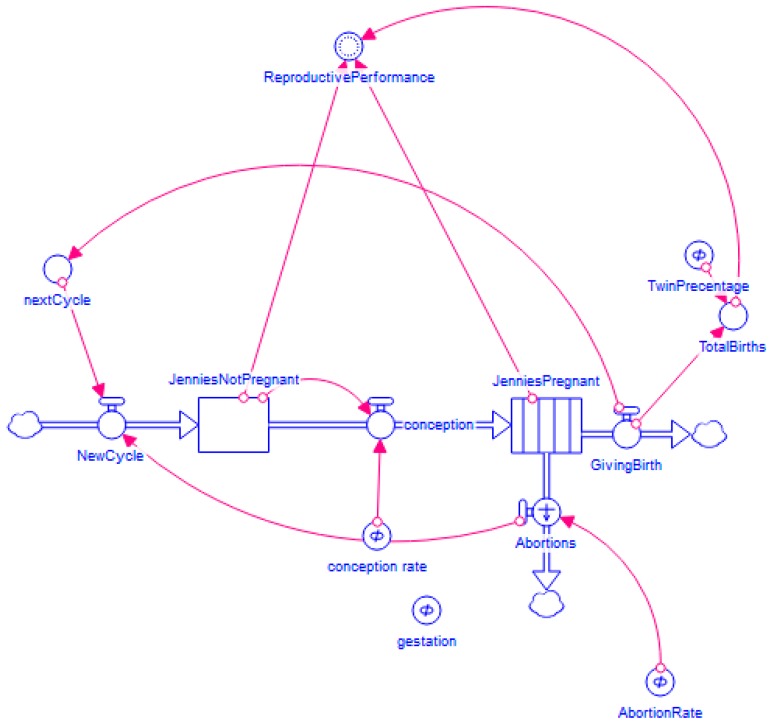
Stock and flow diagram of the reproductive performance model.

**Figure 3 animals-10-00718-f003:**
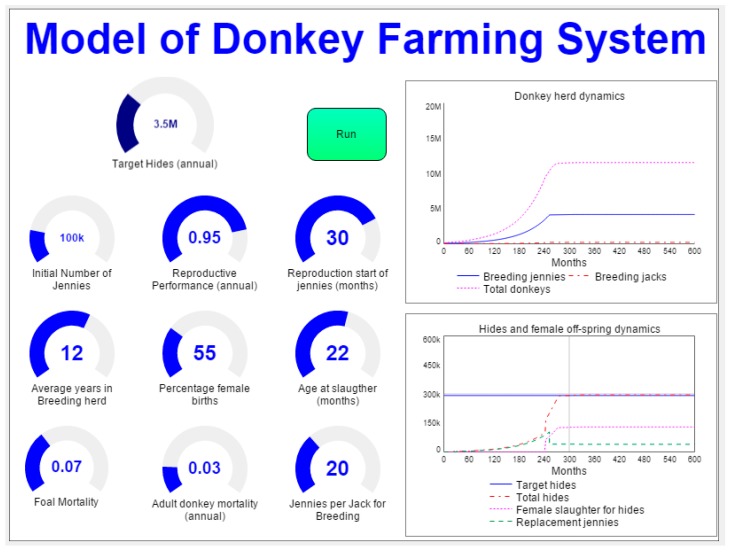
User interface of the donkey farming model.

**Figure 4 animals-10-00718-f004:**
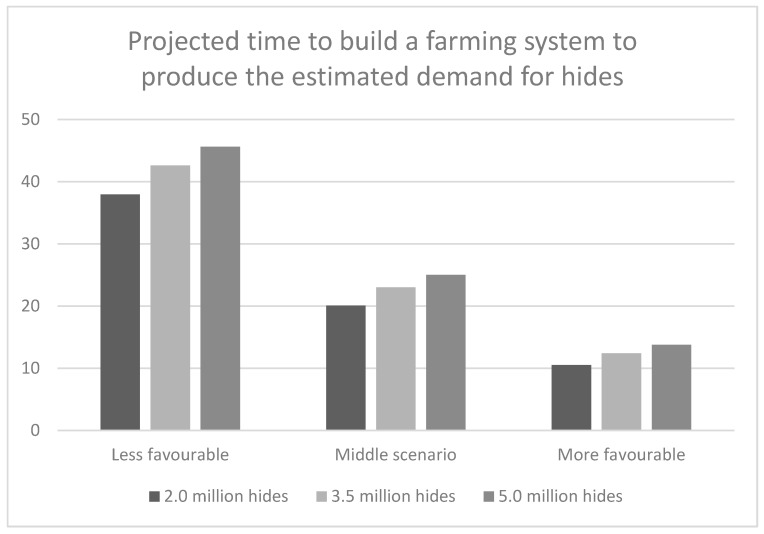
Number of years until the target hides of 2.0, 3.5 and 5.0 million per annum are reached under the three scenarios.

**Figure 5 animals-10-00718-f005:**
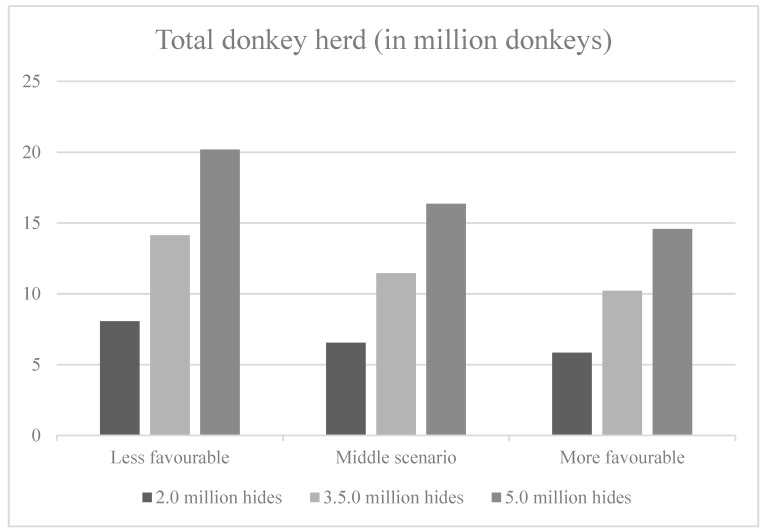
Total donkey herd required for the production of 2.0, 3.5 and 5.0 million hides per annum under the three scenarios.

**Table 1 animals-10-00718-t001:** Summary of the reproductive performance parameters and results.

Variable	Less Favourable Scenario	Middle Scenario	More Favourable Scenario
Conception rate	0.6	0.7	0.8
Gestation length	12.5	12.0	11.5
Abortion rate	0.15	0.10	0.05
Twin percentage	0	5	10
**Results**			
Reproductive performance	0.83	0.95	1.09

**Table 2 animals-10-00718-t002:** Percentage change in reproductive performance compared to middle scenario using less and more favourable values for individual variables.

Variable	Less Favourable Value	% Change in Reproductive Performance	More Favourable Value	% Change in Reproductive Performance
Conception rate	0.6	−1.9%	0.8	1.6%
Gestation length	12.5	−3.8%	11.5	4.2%
Abortion rate	0.15	−3.3%	0.05	3.2%
Twin percentage	0	−4.8%	10	4.8%

**Table 3 animals-10-00718-t003:** Summary of farming model parameter values and results for hide production of 2.0, 3.5 and 5.0 million hides.

Variable	Less Favourable	Middle Scenario	More Favourable
Initial female breeding herd	50,000	100,000	200,000
Reproductive performance	0.83	0.95	1.09
Reproduction age	36 months	30 months	24 months
Average breeding life (annual cull rate)	10 years (10.0%)	12 years (8.3%)	15 years (6.7%)
Percentage female births	50 percent	55 percent	65 percent
Slaughter age	24 months	22 months	17 months
Mortality—foals	10 percent	7 percent	5 percent
Mortality—breeding animals (annual)	4 percent	3 percent	2 percent
Jennie to jack ratio	10	20	30
**Results for 2.0 million hides**			
Hides target reached years	38 years	20 years	11 years
Breeding jennies	2.9 million	2.4 million	2.1 million
Donkey herd	8.1 million	6.5 million	5.8 million
**Results for 3.5 million hides**			
Hides target reached years	43 years	23 years	12 years
Breeding jennies	5.1 million	4.1 million	3.6 million
Donkey herd	14.1 million	11.4 million	10.2 million
**Results for 5.0 million hides**			
Hides target reached years	46 years	25 years	14 years
Breeding jennies	7.3 million	5.9 million	5.1 million
Donkey herd	20.2 million	16.4 million	14.6 million

**Table 4 animals-10-00718-t004:** Percentage change in time to reach target hides, total donkeys and breeding jennies compared to middle scenario using less and more favourable values for individual variables.

Variables (LF, M, MF ^a^)	% Change in Time to Target Hides		% Change in Total Donkeys		% Change in Breeding Jennies	
	LF	MF	LF	MF	LF	MF
Target hides (5, 3.5, 2 million)	9	−13	43	−43	43	−43
Initial female breeding herd (50, 100, 200 thousand)	16	−16	0	0	0	0
Reproductive performance (0.83, 0.95, 1.09)	15	−12	7	−6	15	−13
Reproduction age (36, 30, 24 months)	9	−10	3	−1	1	1
Average breeding life (10, 12, 15 years)	8	−4	1	−1	0	0
Percentage female births (50, 55, 65)	8	−11	0	−1	0	0
Slaughter age (24, 22, 17 months)	1	−1	2	−6	0	0
Mortality—foals (10, 7, 5 percent)	4	−2	2	−2	3	−2
Mortality—breeding animals (annual) (4, 3, 2 percent)	3	−3	2	−1	2	−1
Jennie to jack ratio (10,20,30)	0	0	4	−1	2	0

^a^ LF = less favourable, M = middle, and MF = more favourable.

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
