# Peer review of "The Potential for New Donkey Farming Systems to Supply the Growing Demand for Hides"

_animals, 2020, doi:10.3390/ani10040718_

Round 1

Reviewer 1 Report

Thank you for the opportunity to review this interesting and highly novel piece of research. This work is fascinating and addresses a significant, current global issue facing donkey populations. To my knowledge, this is the most extensive analysis of the economic and demographic landscape surrounding the production of ejiao, and adds significantly to the literature.

I have a few minor comments: 

Line 40                

Remove extra space after “(TCM) and…”              

Line 47                

Please explain why increased industrial and mechanization have played a part. Something like “although increased industrialization and mechanization have certainly played their part, as communities which one relied on donkey traction now use motorised vehicles and farm machinery”.

Line 85

Need a comma after “that in recent years”

Lines 107-110

This paragraph is not necessary as it just repeats what is said above

Line 122

It would be useful to provide the reader with a reference to understand more about Systems dynamics (SD) modelling. For example, could you add a line that says “Full a full description and overview of SD, please see Jones et al. (2001)”.

Line 376

Can you specify “to supply the CURRENT demand for donkey skins” and add a note that in those 10-15 years, demand may be significantly higher, meaning that a farming model may actually never be able to keep up with demand.

Line 387-388

Salmonella abortus equi should be in italics

Line 428

Specify “ejiao production.”

Author Response

Thank you to Reviewer 1 for their detailed and helpful comments and suggested amendments. Each of the suggested changes has been undertaken and shown as track changes in the revised manuscript submitted. In addition to very minor edits and typos the authors have also made the following changes in response to the Reviewer’s comments:

  1. Lines 54-55 provide additional explanation regarding the fall in donkey numbers and increased mechanization.
  2. Formerly lines 107-110 now deleted as suggested.
  3. Lines 135-136 an additional relevant reference with respect to systems dynamics modelling applied to agriculture and resource use has been included (and added to the list of references lines 777-779).
  4. Lines 395-397 added that future demand for ejiao may rise significantly over the next 10-15 years (compared with the current level) and why.
  5. Line 457 “ejiao production” inserted as suggested.

Reviewer 2 Report

This is a very valuable and novel piece of research. 

The introduction overall provides a thorough background and includes all relevant references. It is challenging to find some information relevant to the research topic and authors clearly acknowledge this scarcity of available references containing rigorously data. This paper will hopefully help to highlight the need for more data to be collected and greater transparency.

Some suggestions the authors may consider are:

Within the introduction it is stated that the demand for ejiao has increased and it would be good to include some rationale for why this is the case in recent years (pg 1, lines 41 / 41; pg 2 line 59 / 60) . Additionally, whilst China is a major consumer of this product, it could be good to include if there are trends in ejiao consumption in other countries too and the implications of production and supplying demand to those populations that are consumers living outside of China.

Would it be good to really clearly state in the title, introduction and abstract that despite the focus on China, this issue of notable global relevance due to the current stated supply of donkey hides from Latin America, Australia and African countries?

In the introduction sourcing of donkey hides is discussed without reference to any specific companies. Does it need to be stated that the source, sink / supply chains and point of sale are very fragmented or are there some key companies that have oversight of the sourcing and sale and if so, do they have clear and transparent souring policies?

It may be good to make clear in the introduction the model being is based purely on donkey hide demand and not inclusive of donkey milk and meat demand.

It is noticeable that in both the introduction and discussion that the implications for the welfare of donkeys in the current and future hide supply chains and farming systems are not explored in depth in the paper. Or mention of how welfare standards can / could be regulated and assured. Also, any potential ecological and conservation issues. There is the possibility that the readership of the journal ‘Animals’ may be interested in these perspectives, however.

In the Materials and Methods section 2.1. Approach it could be good to reiterate that the model under development is based purely on donkey hide demand and not inclusive of donkey milk and meat demand. In section 2.2 Reproductive Performance Model, the reason for having the number of 1000 jennies as the population could be rationalised (line 151). Section 2.3. The Donkey Farming System Model is very well researched and discussed.

The discussion is very comprehensive. In relation to the wider implications of this research, the World Health Organisation and World Health Assembly have a traditional medicine strategy and it may be worth acknowledging this in the paper and the implications ( https://www.who.int/medicines/publications/traditional/trm_strategy14_23/en/ ). Could some reference to the need to identify alternative herbal medicines to supply an equivalent demand and therapeutic effect? Also how is this story that is being told about ejiao fitting in with various global activities, such as governments (including China) having signed support and agreement towards working towards achievement of the United Nations Sustainable Development Goals by 2030, and this being more on key companies radars also. How might what the model reveals about donkey farming systems and potential supply of ejiao cross relate to achievement of SDGs and WHO? 

Overall the paper addresses a very complex and under-researched issue and uses elaborate research approach and analysis to quantify the issue.

Author Response

Thank you to Reviewer 2 for their helpful comments and suggested amendments. Each of the suggested changes has been undertaken and shown as track changes in the revised manuscript submitted. These changes are outlined below.

  1. Line 41 makes the point that it is demand for ejiao in TCM worldwide not just in China (although China is the largest market of course).
  2. Lines 42-43 provide reasons for why demand for ejiao has increased.
  3. It is now made clear in the Simple Summary, Abstract and Introduction (see addition of lines 91-93 in the latter) that the issue is a global one.
  4. Lines 46-51 make reference to a specific company in China which is considered the market leader and its sourcing policy.
  5. The point is now made that the model considers production for hides and not for milk, meat or other donkey-derived products (see new lines 94-95 and line 126 at the start of Materials and Methods).
  6. Additional comment has been made in relation to donkey welfare, welfare standards and policy on lines 412-422 as suggested.
  7. The reason for having 1000 jennies as a population starting point in the model is provided on lines 164-166.
  8. Lines 399-403 have been added to acknowledge the WHO World Health Assembly ICD 11 inclusion of a chapter on TCM and the influence and possible implications of this. The relationship of ejiao production to SDGs is a very complicated one with potential positives and negatives but also a lack of evidence for the authors to meaningfully comment.

Reviewer 3 Report

The genesis of this paper is that the demand for donkey hides for gelatin used in Traditional Chinese Medicine (ejiao) has greatly increased in China in recent years, with the result that prices for donkey hides are rapidly increased. As a consequence the authors explore the feasibility of setting up new donkey farming systems to supply the rising demand for ejiao in China using a systems dynamics model of donkey production.

As such this article represents an excellent study.

The objectives of the paper are of interest and fit well within the scope of the journal.

The work appears well conducted and the results reported are pertinent.

The language is fluent and appropriate.

In my opinion, the manuscript could be accepted for publication in Animals.

Author Response

Many thanks to the Reviewer for their comments.